# The Challenging Approach to Multiple Myeloma: From Disease Diagnosis and Monitoring to Complications Management

**DOI:** 10.3390/cancers16122263

**Published:** 2024-06-19

**Authors:** Sonia Morè, Laura Corvatta, Valentina Maria Manieri, Erika Morsia, Massimo Offidani

**Affiliations:** 1Clinica di Ematologia Azienda Ospedaliero, Universitaria delle Marche, 60126 Ancona, Italy; s.more@staff.univpm.it (S.M.); valentina.manieri95@gmail.com (V.M.M.); e.morsia@staff.univpm.it (E.M.); 2U.O.C. Medicina, Ospedale Profili, 60044 Fabriano, Italy; laura.corvatta@sanita.marche.it

**Keywords:** multiple myeloma, bispecific antibodies, CAR T cells, infection, side effects

## Abstract

**Simple Summary:**

Multiple myeloma (MM) represents the second most common hematological malignancy, but its diagnosis can be significantly delayed since symptoms are not specific and, mainly in the older population, alternate diagnoses can mimic MM. Bone marrow biopsy (evaluating the amount of proliferating myeloma cells) remains an essential procedure, but several imaging methods such as whole-body low-dose computed tomography, positron emission tomography or whole-body magnetic resonance have become crucial for the diagnosis and staging of MM, and are also taking on a prognostic role. MM is a clinically and biologically heterogeneous disease; therefore, with the aim to identify patients with different outcomes, some risk models such as ISS, R-ISS or R2-ISS have been proposed over time. However, the most recent attempts have been to establish individualized patient risk, integrating clinical, genomic and therapeutic data in order to personalize treatment and avoid overtreatment and toxicities.

**Abstract:**

The outcome of multiple myeloma (MM) has significantly improved in the last few decades due to several factors such as new biological discoveries allowing to better stratify disease risk, development of more effective therapies and better management of side effects related to them. However, handling all these aspects requires an interdisciplinary approach involving multiple knowledge and collaboration of different specialists. The hematologist, faced with a patient with MM, must not only choose a treatment according to patient and disease characteristics but must also know when therapy needs to be started and how to monitor it during and after treatment. Moreover, he must deal not only with organ issues related to MM such as bone disease, renal failure or neurological disease but also with adverse events, often very serious, related to novel therapies, particularly new generation immunotherapies such as CAR T cell therapy and bispecific antibodies. In this review, we provide an overview on the newer MM diagnostic and monitoring strategies and on the main side effects of MM therapies, focusing on adverse events occurring during treatment with CAR T cells and bispecific antibodies.

## 1. Introduction

In patients with multiple myeloma (MM), accounting for 10% of all hematological malignancies, the clinical features of the disease and the potential complications caused from therapies require close collaboration between hematologists and different specialists, starting with laboratory assistants, nephrologists, radiologists, neurologists, sometimes neurosurgeons and others. This malignancy is characterized by the uncontrolled growth of clonal plasma cells, which, accumulating in the bone marrow and producing M-protein, cause lytic bone lesions, renal injury, anemia and hypercalcemia, defining the so-called CRAB criteria. However, the occurrence of extramedullary disease (EMD), excluding paraskeletal extramedullary disease, diagnosed in up to 13% of MM patients (7% at diagnosis and 6% during follow-up) [1] can induce a wide spectrum of site-specific symptoms resulting in diagnostic challenges. Although the biological heterogeneity of MM is defined and stratified mainly according to FISH analysis, this method is not able to capture the molecular complexity of the disease. Novel technologies such as whole-genome sequencing, whole exome sequencing and targeted panels go so far as to detect heterogeneity within the single molecular subgroup, allowing not only to improve risk stratification [2] but also to find targets for new therapies [3] and to identify genetic features related to drug resistance [4].

The introduction of agents as proteasome inhibitors (PIs) and immunomodulatory agents (IMiDs) and more recently of monoclonal antibodies (mAbs) as daratumumab and isatuximab has resulted in significant improvements in progression free survival (PFS) and overall survival (OS) either in newly diagnosed (NDMM) or in relapsed/refractory MM (RRMM). Although a longer follow-up is needed to establish long-term effectiveness of new generation immunotherapies such as CAR T cell therapy and bispecific antibodies, they have shown to induce unprecedented rates of deep response in patients with very advanced MM. However, when using these novel immunotherapies, clinicians have discovered side effects not reported for previous treatments such as cytokine release syndrome (CRS) and neurotoxicity, including immune effector cell-associated neurotoxicity syndrome (ICANS), with high rates of life-threatening infections [5]. In this review, we summarize the most recent topics on MM diagnostic and monitoring and we describe the most frequent side effects occurring during standard therapies, focusing on those developed during treatment with CAR T cells and bispecific antibodies.

## 2. Latest in Diagnosis: From X-rays to Artificial Intelligence-Based Techniques

Criteria for the diagnosis of MM, the term used for a disease requiring therapy, were updated by the International Myeloma Working Group (IMWG) in 2014, adding to the traditional CRAB criteria (hypercalcemia, renal failure, anemia, at least one lytic bone lesion) and SLiM criteria, including 60% of clonal bone marrow plasma cells, involved/uninvolved serum free light chain ratio ≥100 and >1 focal lesion in MRI studies [6]. Among imaging techniques, conventional X-ray examination is no longer considered a gold standard due to the low sensitivity to detect bone lesions (false negativity ranging from 30% to 70%) and the inability to detect extramedullary disease [7]. The most recent consensus by IMWG [8] recommended that, in the suspected MM, the patient undergoes low-dose whole-body CT (WBLDCT) or, if not available, a PET/CT scan. The first method, characterized by a radiation dose notably lower than that of conventional CT [9] and shorter acquisition times, can be performed from the cranium to the proximal tibia metaphysis, showing sensitivity and specificity of 70% and 90%, respectively, and allowing to detect extramedullary lesions. The emerging photon-counting detector CT technology (DS-PCD-CT) might represent a resource in the future for diagnosing MM bone disease, since it improves the visualization of fine details, generates multi-energy images with up to 47% lower image noise with reduced radiation exposure and, in contrast with the current energy integrating detector (EID) CT, allows to assess the qualitative and quantitative composition of bone marrow [10]. The recent study by Winkelmann et al. was the first to demonstrate a significant superiority of DS-PCD-CT vs. DS-EID-CT in terms of spatial resolution of bony microstructure and lytic bone lesions in 50 patients with MM [11]. PET/CT imaging, using a radiolabeled glucose analog FDG, visualizes bone lesions and extramedullary disease, by abnormal glucose metabolism, with sensitivity and specificity ranging from 80 to 100% [12]. The semiquantitative analysis, utilized for evaluating metabolic behavior, is expressed in terms of standardized uptake value (SUV), which is a mathematically derived ratio of tissue radioactivity concentration at a point in time C (T) at a specific region of interest (ROI) and the injected dose of radioactivity per kilogram of the patient’s body weight. Several factors can affect SUV measurement, including blood glucose levels, the patient’s weight, the ROI of interest and the size of the matrix; remarkably, there is not currently an SUV value that is considered normal [7]. Moreover, a false positivity of PET/CT imaging can be found in the presence of infections, inflammations, bone remodeling or bone marrow repopulation after therapy [12]. On the other hand, Rasche et al. found an incidence of PET/CT false negativity in 11% of newly diagnosed MM patients and this phenomenon was associated with a low expression of gene coding for herokinase-2, catalyzing the first step of glycolysis [13]. Going back to the IMWG consensus, a whole-body MRI (WB-MRI) (or MRI of the spine and the pelvis if the first is not available) is recommended when WBLDCT or PET/CT are negative. Whole-body MRI (WB-MRI) represents the most sensitive and specific technique to detect the different patterns of bone marrow infiltration (focal, micronodular, diffused or mixed pattern) before the appearance of osteolytic lesions [14] and it is able to assess spinal canal and nerve root compression causing neurological complications. In contrast with CT or PET/CT, it offers the benefit of not causing radiation exposure, not requiring a pre-scan diet and not being reliant on the metabolic activity of tumor cells [15]. In a prospective observational study comparing baseline WB-MRI and FDG PET/CT and including 60 patients with newly diagnosed or relapsed/refractory MM, at least one focal lesion was detected in 60% and 83% of cases using PET/CT and WB-MRI, respectively, demonstrating that this latter imaging technique showed a significantly higher detection of focal lesions at all anatomic sites except ribs, scapulae and clavicles [16]. Moreover, plasma cell infiltration and serum paraprotein levels were significantly higher in diffuse disease detected at WB-MRI compared with no diffuse disease; whereas, these correlations were not found with FDG PET/CT [16]. Diagnosis of MM requires either clonal bone marrow plasma cells of 10% or more (detected by conventional aspiration or biopsy of iliac crest) or a biopsy-proven plasmacytomas [6] to differentiate gammopathy of undetermined significance (MGUS) from smoldering multiple myeloma (SMM) and MM, although a large retrospective study by Mayo Clinic showed that omitting a bone marrow biopsy in patients with low-risk MGUS and without CRAB features or a markedly elevated ratio of involved-to-uninvolved FLC resulted in missed diagnosis of MM or SMM in <1% of patients [17]. Recently, dual-energy computed tomography (DECT), by creation of virtual non-calcium images (VNCa) and support of artificial intelligence (IA), was found to be able to identify MM patients without osteolytic lesions on conventional CT with a sensitivity and specificity of 0.63 and 0.71, respectively. Moreover, this technique allowed to select patients with higher pre-test probability of myeloma, defining bone lesions and clinical diagnosis of MM [18]. In the study by Xiong et al., the DECT technique exhibited a satisfactory performance for discriminating high and low serum FLC ratios and for the prediction of high risk vs. standard cytogenetic status [19].

Although the updated IMWG criteria added SLiM diagnostic criteria [6], a recent meta-analysis evaluating whether the risk of progression of MM patients with SLiM CRAB criteria to classical CRAB criteria described in studies published after 2014 differed from that reported in studies leading to new criteria published in 2014, showed that, in studies published after 2014, patients with ≥60% bone marrow plasma cells or an FLC ratio ≥100 had an approximately three times longer TTP and 50% lower 2 yr progression risk compared with patients included in the earlier studies, suggesting that MM patients with the above SLiM findings may not be treated immediately but be carefully followed up by stringent laboratory tests and medical examinations [20]. However, in patients with an FLC ratio ≥100, the 2-year risk of progression to symptomatic MM (typical CRAB criteria) was significantly higher in the presence of urine monoclonal protein excretion ≥200 mg/24 h vs. <200 mg/24 h (36.2% vs. 13.5%, respectively, *p* < 0.001) [21]. The ongoing prospective IFM-2017-04 (CARRISMM) study (NCT04144387) is evaluating the impact of the updated MM criteria on the natural history of SMM in order to establish new recommendations about follow-up and prognostication of SMM.

## 3. From More Precise MM Risk Stratification to Personalized Therapy

The extreme inter-patient and intra-patient heterogeneity of MM, responsible for the significant variability in outcome, represents a challenge to building a risk stratification model. The most recent and commonly used systems such as the revised international staging system (R-ISS) and R2-ISS have integrated new elements into the traditional ISS staging system, based on the measurement of serum β2-microglobulin and albumin, representative of disease burden [22]. R-ISS incorporates high-risk chromosomal abnormalities such as del(17p), t(4; 14), t(14; 16) assessed by FISH analysis and serum lactate dehidrogenase (LDH) [23]. However, according to this model, a patient is classified as middle risk (R-ISS stage II) with or without high-risk cytogenetic abnormalities, making this group of patients very heterogeneous regarding outcome. In a recent retrospective analysis of 1614 transplant eligible NDMM patients, IFM showed that those with R-ISS II could be stratified into three subgroups, according to ISS stage and chromosomal abnormalities, with significantly different OS [24]. The more recent R2-ISS [25] included, among cytogenetic abnormalities, 1q21 abnormalities, grouping together gain (3 copies) and amplification (>3 copies), but not t(14; 16) whose value is still being debated. If some authors report no prognostic impact of isolated t(14; 16) [26] or suggest that only its interaction with other high-risk abnormalities is associated with worse outcome [27], others peremptorily state that, being a primary translocation event, t(14; 16) remains one of the markers with the greatest impact on outcome [28]. Moreover, R2-ISS stage system did not include del(1p32), which, in a large cohort of NDMM, was found to significantly affect OS, being 49 months vs. 124 months in those displaying or not this abnormality, respectively. Notably, patients with worse outcome were those with biallelic del(1p32), since the median OS was 25 months vs. 60 months in those with monoallelic del (1p32) [29]. The development of next generation sequencing (NGS) technologies has allowed to identify genetic markers with prognostic and potentially therapeutic values. Walker et al. [30] identified double-hit patients with extremely poor outcome (median PFS 15,4 months and median OS 20.7 months) characterized by either biallelic TP53 alterations or amplification of 1q21 and ISS stage III. Using whole-exome sequencing (WES) and whole-genome sequencing (WGS) studies, some mutational signatures have been found to show a prognostic value such as the APOBEC mutational signature, linked to a high mutational load and the adverse t(14; 16) and t(14; 20) MAF-translocation subgroups [31,32]. Recently, the MMRF CoMMpass dataset has been used to develop a risk score (defined as the editor/inflammation score (EI-score)), which, incorporating mRNA levels of APOBEC genes, pro- and anti-inflammatory genes and clinical markers such as β2-microglobuline and LDH, was able to improve the performance of ISS, R-ISS and R2-ISS [33]. A significant breakthrough in the prognostic stratification of MM was achieved with the help of artificial intelligence (AI) and deep neuronal networks by Maura et al. [2], who developed a model able to predict individualized risk in MM (IRMMa), integrating clinical, genomic such as 1q21 gain/amplification, del1p, TP53 loss, NSD2 translocations, APOBEC mutational signature, copy-number signatures and therapeutic data of 1933 patients with NDMM. The IRMMa accuracy was significantly higher than that of other models, being the c-index for OS 0.726 vs. 0.61, 0.572 and 0.625 for ISS, R-ISS and R2-ISS models, respectively. The novelty of this model, compared with the others, was that, across the 12 identified genomic groups, the risk was affected by the type of treatment (treatment variance), allowing to identify, for example, patients benefiting from immediate ASCT. Developments in clonal and subclonal knowledge of MM will probably lead to radical changes in the risk stratification of this hematologic disease. In a recent consensus meeting [34], patients at high risk were defined as follows: presence of t(14; 16) due to aberrant APOBEC activity; clonal fraction of del17p higher than 20% or biallelic inactivation of P53, mutation or loss of copy of P53; 1q gain or del1p with ISS stage III; whereas, t(4; 14) was omitted from the high-risk definition. In addition to genetic features, some clinical presentations are associated with poor outcome such as extramedullary MM (EMM) [35]. In the recently published NGS study of EMD tumor cells [36], 79% of EMM samples were characterized by co-occurrence of 1q21 gain/amplification and MAPK pathway mutations, and NDMM patients with mutated KRAS and1q21 gain/amplification had a significantly higher risk of EMM occurrence (HR = 2.3, *p* = 0.011). However, the most intriguing finding was decreased expressions of CD38, SLAMF7, GPRC5D, FCRH5 and MHC-I (required for recognition of plasma cells by T cells) on EMM cells, supporting the reported low efficacy of daratumumab, bispecific antibodies and CAR T cells in this setting. On the other hand, in the EMM cell upregulation of EZH2 and CD70, possible new therapeutic targets and a microenvironment mainly including CD8+ T cells and NK cells were observed.

Also, circulating plasma cells (CTCs) assessed by next generation flow cytometry (NGF) have been found to be a factor significantly affecting outcomes [37,38]. Among transplant eligible NDMM patients, 3-year PFS was 87% vs. 60% in patients without and with CTCs (HR = 0.20, *p* < 0.001), whereas 3-year OS was 95% vs. 80%, respectively (HR = 0.19, *p* = 0.002). Notably, high-risk cytogenetic abnormalities had no impact on PFS and OS of NDMM patients without CTCs [39]. A recent meta-analysis [40], including 22 studies with a total of 5637 patients, confirmed the prognostic value of CTCs, since it demonstrated that patients with elevated CTCs level, with different thresholds between studies, were expected to have a poor PFS (HR = 2.45; *p* < 0.001) and OS (HR = 2.19; *p* < 0.001).

Imaging techniques can also be used for prognostic stratification, as shown by Rasche et al. [41], who reported significantly poor PFS and OS in patients with at least three large focal lesions (> 5 cm^2^) on MRI. This finding, observed in 13.8% of patients, was independent of R-ISS, gene expression profiling (GEP)-based risk score, gain 1q or EMM. MRI using three-dimensional (3D) convolutional neural networks (CNNs) and AI [42] or whole-body volumetric calculation of bone marrow metabolism after application of an automated deep learning-based tool on PET/CT images [43] could represent in the future further methods to assess the risk of MM patients.

Longitudinal studies of MM genome sequenced at diagnosis and at subsequent relapses showed the acquisition of novel abnormalities associated with drug resistance. In a large whole-genome sequencing data set of RRMM, Ansari-Pour et al. [4] demonstrated that TP53, DIOX2, 1qgain and 17p loss-of-heterozygosity increased from NDMM to lenalidomide resistant to pomalidomide resistant stages, these features being drivers of relapse and conferring clonal selective advantage. Durand et al. [44] established a functional p53 score able to identify MM cells with biallelic TP53 invalidation and found that specific inhibitors of anti-apoptotic BCL2 proteins (i.e., BH3 mimetics) could be of interest in MM with this high-risk feature, since p53-regulated BAX is critical for optimal response to BH3 mimetics. Recently, an association between high CCR1 (C-C motif chemokine receptor 1) and bortezomib resistance in MM cell lines was found [45]. CCR1, an independent poor prognostic factor in NDMM receiving bortezomib-based regimens [46], is upregulated by transcription factors MAF and MAFB, whose expression is related to t(14; 16) and t(14; 20). CCR1 may represent a therapeutic target whose inhibition could restore sensitivity to bortezomib. Another recent study [47] identified a novel biomarker and therapeutic target ribosomal protein S3 (RPS3), a component of the 40S subunit of the eukaryotic ribosome, which is overexpressed in MM and associated with poor prognosis. Notably, the overexpression of RPS3 promotes MM proliferation and confers proteasome inhibitor resistance to MM, whereas the knockout of RPS3 induces MM cell apoptosis. The first-in-class selective inhibitor of ribosome biogenesis CX-5461 (Pidnarulex), by topoisomerase II trapping and replication-dependent DNA damage leading to cell death, was found to exert potent antimyeloma activity in PI-resistant MM preclinical models [48]. By integrated analysis of WGS and characterization of microenvironments of patients treated with quadruplet daratumumab, carfilzomib, lenalidomide and dexamethasone (D-KRd), Maura et al. [49] found several genomic drivers and microenvironment features associated with drug resistance, response and duration of response, suggesting the possibility of robust prediction models to personalize treatment. Recently, a targeted sequencing panel (myeloma genome project panel, MGPP), capturing 283 loci recurrently mutated in MM, was evaluated to see if it could be useful both to better identify high-risk patients and be a treatment decision support [50]. Overall, among 46 NDMM patients undergoing FISH analysis and MGPP at baseline, 13 (28%) were reclassified as high risk with MGPP vs. FISH, since they were enriched for prognostically significant variants such as MYC rearrangement, subclonal 17p loss, 1q gain and 1p loss. Remarkably, based on this altered risk designation, 24% of patients were reclassified to receive intensified treatment and 12% tandem ASCT. Moreover, MGPP identified eligible patients for targeted therapy such as MEK/RAF inhibitors (46%), CDK inhibitors (15%) or venetoclax (13%). However, unlike solid tumors, the molecularly oriented precision medicine approach does not seem to be more effective than conventional antimyeloma therapy, as reported in the recent MM-EP1 study, probably due to the limited number of available targeting agents and to the heterogeneity of tumor cells requiring combined targeted therapies [51].

## 4. Disease Monitoring at the Time of Increasingly Effective Therapies

Awaiting the personalization of the future treatment of MM, based mainly on the genetic characteristics of disease, the introduction of triplet or quadruplet combinations, including PIs, IMiDs and mAbs in the upfront setting, and the development of new generation immunotherapies such as bispecific antibodies and CAR T cells, allows to make deeper and longer lasting but not infinite responses to therapy. Therefore, the need for new definitions of response is increasing over time, since, although treated with increasingly effective therapies (Table 1), MM patients relapse after a longer or shorter time. Among several studies demonstrating the prognostic role of measurable residual disease (MRD), the meta-analysis by Munshi et al. [52] showed that MRD status after treatment predicted PFS and OS in all MM patients, including those obtaining a CR whose median PFS and OS were 34 and 82 months, respectively, if MRD was negative, and 56 months and 112 months, if MRD was negative (*p* significant for both PFS and OS). Based on these data, IMWG updated response criteria, adding the new categories of MRD negativity (flow MRD-negative, sequencing MRD-negative and imaging-positive MRD-negative) and sustained MRD-negativity defined as MRD negativity in the marrow and by imaging confirmed minimum of 1 year apart [53]. As recommended by IMWG, the two most appropriate methods to detect bone marrow MRD are flow cytometry (NGF) and high-throughput DNA sequencing (NGS). NGF, using 2-tube 8-color methodology [53] or single-tube 10- or 12-color methodology [54], achieves a sensibility between 10-5 and 10-6 and does not require a sample at diagnosis [55]. In contrast, NGS, able to reach a sensitivity of 10-6, requires a baseline bone marrow sample to identify the predominant clone that can be detected in almost 90% of MM patients [56]. In a prospective MRD analysis of the Phase II FORTE trial [57], enrolled transplant eligible NDMM, NGF and NGS were concordant in 87% and 83% analyses at the 10-5 and 10-6 cut-off, respectively, translating into a clinical and prognostic concordance, being HRs in NGF-MRD and NGS-MRD negative vs. positive patients of 0.29 and 0.27 for PFS and 0.35 and 0.31 for OS, respectively (*p*< 0.05). In the recent meta-analysis by Munshi et al. [58], MRD negativity was associated with significantly longer PFS (HR = 0.33, *p* < 0.001) and OS (HR = 0.45, *p* < 0.001), either in NDMM or RRMM patients, regardless of methods used or sensitivity thresholds; although, the best effect on outcome was observed with an MRD negativity level of 10-6. Cavo et al. [59] confirmed these data and, in a pooled analysis of four Phase III trials (POLLUX, CASTOR, ALCYONE and MAIA) evaluating daratumumab in combination with standard-of-care regimens for NDMM and RRMM, showed that patients achieving at least CR with MRD negativity had significantly improved PFS, irrespective of therapy or disease setting. Currently, it seems to be taking on increasing importance not only to achieve MRD negativity but also to maintain it over time. Patients with a sustained MRD negativity lasting ≥12 months showed a significant improved PFS compared with those who were MRD positive or with MRD negativity lasting less than 12 months [60].

Predictors of sustained MRD negativity were analyzed in two recent studies by Italian and Spanish groups. D’Agostino et al. [61], evaluating MRD status every 6 months after the start of maintenance (lenalidomide with or without carfilzomib) in patients enrolled in the FORTE trial, found, as factors associated with a higher risk of MRD recurrence, baseline high risk cytogenetic abnormalities, circulating tumor cells and time to reach MRD negativity (post-consolidation vs. pre-consolidation). Guerrero et al. [62], again in transplant eligible NDMM, achieved similar results, since risk factors for MRD resurgence were ISS 3, circulating tumor cells and failure to obtain MRD negativity after induction therapy. Also, RRMM patients receiving new immunotherapies as CAR T cells and bispecific antibodies, sustained MRD negativity affected PFS and OS but, in contrast with NDMM, achieving CR did matter in MRD negative RRMM patients with respect to response durability [63]. Other than unresolved issues regarding the most appropriate sensitivity or the optimal time for MRD assessment, it has to be emphasized that current MRD measurement requires repeated invasive procedures. So, studies are ongoing to understand whether bone marrow-based methods can be replaced by less invasive techniques in order to implement MRD assessment in clinical practice. Using BloodFlow, a highly sensitive method (10-7) combining immunomagnetic enrichment with NGF, Gonzalez et al. [64] showed a 12-fold increment in the risk of progression and/or death in MRD positive patients who were on maintenance or observation in the PETHEMA/GEM clinical trial. Moreover, the concordance between BloodFlow in blood and NGF in bone marrow was 79%, and double-negative detection using BloodFlow achieved a negative predictive value (NPV) of 78%, being a positive predictive value (PPV) of 96%. Another minimally invasive method to assess MRD in peripheral blood is mass spectrometry (MS), allowing the sensitive detection of M-proteins based on the unique sequence of the antigen binding region (CDR) of Ig [65]. Data from clinical trials enrolling transplant-eligible NDMM in the STAMINA [66], GMMG-M5 [67] and ATLAS trials [68], MS (MALDI-TOF MS) showed a significant negative impact of MS positivity on PFS also in CR patients, and agreement between MS and other methods and NGF and NGS increased with time [68]. However, prospective studies are needed to better define the role and appropriate timing of MS for MRD assessment (some are ongoing (NCT05536700, NCT06189833, NCT05686447)) [69]. Imaging plays a key role in MRD evaluation, mainly PET/CT, whose prognostic value has been demonstrated in several studies [70,71]. Recently, using the Deauville scores (DS) to define PET complete metabolic response (CMR) in patients enrolled in the Phase II FORTE trial, PET negativity was found to significantly affect PFS, although the best outcome was observed in patients achieving before maintenance both PET/CT CMR and NGF negativity at level 10^−5^ [72]. Sustained WB-RI negativity also, according to MY-RADS criteria [73], predicted outcome, since patients on maintenance after ASCT who were imaging MRD negative at 1 year (RAC 1) had a significantly longer PFS (median 55.4 vs. 28.4 months; HR = 0.12; *p* < 0.0001) and OS (median NR vs. 63 months; HR = 0.13; *p* < 0.0007) compared with patients with residual disease on WB-MRA (RAC ≥ 2) [74].

In Table 2, we summarize the current and potential future laboratory tests and imaging methods for the diagnosis and monitoring of MM in the clinical practice.

**Table 1 cancers-16-02263-t001:** MRD rates in most recent regimens for eligible and not eligible for ASCT newly diagnosed MM patients.

Trial	Phase	Treatment	MRD Negativity (%)/Sensitivity 10^−5^	Timing of MRD Assessment
**ASCT eligible**				
CASSIOPEIA [75]	III	Dara-VTD (4) → ASCT → Dara-VTD (2)	64	Post-consolidation
GRIFFIN [76]	II	Dara-VRD (4) → ASCT → Dara-VRD (2)	50	Post-consolidation
PERSEUS [77]	III	Dara-VRD (4) → ASCT → Dara-VRD (2)	57.5	Post-consolidation
MASTER [78]	II	Dara-KRD (4) → ASCT → Dara-KRD (4)	81	Any time ^^
IFM 2018-04 * [79]	II	Dara-KRD (6) → ASCT → Dara-KRD (4) → ASCT	97	After second ASCT
GMMG-HD7 [80]	III	Isa-VRD (3) → ASCT	50.1	Post-induction
IsKia EMN 24 [81]	III	Isa-KRD (4) → ASCT → Isa-KRD (4)	77	Post-consolidation
GMMG-CONCEPT ^ [82]		Isa-KRD (6) → ASCT → Isa-KRD (4)	67.7	Post-consolidation
**ASCT not eligible**				
MAIA [60]	III	Dara-Rd continuously	28.8	Any time **
ALCYONE [60]	III	Dara-VMP × 9 cycles	26.9	Any time

* Patients at high risk with at least one high-risk cytogenetic abnormality among del(17p), t(4; 14) or t(14; 16); ^ Patients at high risk with ISS stage II/III combined with (17p), t(4; 14) or t(14; 16) or more than 3 1q21 copies. ^^ MRD status was assessed after induction, after ASCT, after 2 cycles of Dara-KRD as consolidation and at completion of consolidation. ** MRD status was assessed in all patients who achieved ≥ CR.

**Table 2 cancers-16-02263-t002:** Current and potential future tests for diagnosing and monitoring of multiple myeloma.

Diagnosis	Risk Stratification	Monitoring (MRD Assessment)
**Laboratory tests** Complete blood count, differential and platelet count; Blood urea nitrogen (BUN), cretinine, creatinine clearance, electrolytes, calcium, liver function tests, albumin, LDH, β2-microglobulin; Serum protein electrophoresis (SPEP) and serum immunofixation electrophoresis (SIFE); Serum free light chain (FLC) assay; NT-proBNP (or BNP); 24 h urine for total protein, urine protein electrophoresis (UPEP) and urine immunofixation electrophoresis (UIFE); Circulating plasma cells. **Bone marow analysis** Bone marrow aspirate and biopsy, including immunohistochemistry; Possible storage of aspirate sample for future MRD testing by NGS. **FISH analysis** Panel including de17p, t(4; 14); t(14; 16), t(14; 20), 1q21 gain/1q21 amplification, 1p delection **Molecular analysis** TP53 mutation/deletion **Imaging methods** Whole-body low-dose CT or FDG-PET/CT; If whole-body low-dose CT or FDG-PET/CT negative, whole-body MRI or MRI of spine and pelvis. **Other potential future methods** Dual-energy computed tomography (DECT) [18,19].	ISS [22]; R-ISS [23]; R2-ISS [25]. **Potential future risk scores** EI Score [33]; IRMMa Score [2]; Barcelona Score [34]; Whole-body MRI with artificial intelligence [42]; PET/CT with artificial intelligence [43].	Next generation flow cytometry (NGF); Next generation sequencing (NGS); PET/CT (Deauville scores) [72]; Whole-body MRI (MY-RADS criteria) [73]. **Potential future methods** Blood flow [64]; Mass spectrometry (MS) [65].

## 5. New Treatment and Novel Side Effects

The management of novel treatment side effects could be one of the most relevant interdisciplinary settings for MM patients in the novel therapeutic era. Myelosuppression, thromboembolic events, peripheral neuropathy, steroid toxicities and gastrointestinal side effects are already well known by clinicians, but novel immunotherapies have brought new toxicities, needing data and longer follow-ups to be better characterized. BCMA- and GPRC5D-targeted immunotherapies, both bispecific antibodies and CAR T cells, could potentially cause early toxicities, given by exaggerated immune responses related to T cell activation and proliferation and consequent cytokine storms and late or delayed toxicities, developing in the months or years after the start of therapy, respectively [83]. Cytokine release syndrome (CRS) is one of the early complications, clinically manifesting with fever, respiratory failure and/or hypotension, until shock and multiorgan failure manifest in more severe cases, usually accompanied by changes in laboratory parameters, including elevated C-reactive protein, ferritin, lactate dehydrogenase and coagulation labs [84]. It is usually limited to step-up doses and first full dose and occurs at a median of 1 to 2 days after the dose. In the setting of bispecific antibodies, the all grades CRS rate is about 50–70%, with grade ≥3 CRS lower than 5% (teclistamab 72%/0.6%, elranatamab 57%/0, REGN5458 50%/0.6%, alnuctamab 53%/0, ABBV-383 70%/<5%), with differences based on dosing [85,86,87,88,89]. Most patients treated with CAR T cells experience CRS, although grade 3 or higher are less common (ide-cel 84%/5%, cilta-cel 95%/4%) [90,91]. The management of CRS deals with a correct differential diagnosis with other possible causes of fever, mostly infections, and an accurate treatment based on its grading and severity, according to the American Society for Transplantation and Cellular Therapy recommendations [92]. Rajeeve et al. presented preliminary results of a trial to early and consistently identify CRS using a wearable device for remote patient monitoring in 34 RRMM patients treated with CAR T cells. The goal is to identify device data that, alone or in combination with the analysis of trends in cytokine biomarkers, help the early CRS detection in outpatient RRMM, optimizing their treatment [93]. Disease burden being the strongest predictor for CRS, bridging chemotherapy remains the best mitigation strategy before lymphodepletion [93,94]. Prophylactic strategies do not exist, but studies are ongoing regarding tocilizumab and anakinra before bispecific antibodies, demonstrating a significant reduction in the CRS rate if used pre-cevostamab administration [95,96,97,98,99]. Immunological effector cell-associated neurotoxicity syndrome (ICANS) is a less common early toxicity, whose underlying driving mechanism is not fully understood, but the release of inflammatory cytokines secreted by macrophages and monocytes, increasing vascular permeability and endothelial activation and leading to a blood–brain barrier breakdown, seems to be the principal pathogenic mechanism. ICANS is a clinical diagnosis, characterized by a broad spectrum of neurological symptoms such as mild tremor and confusion, which can then proceed to agitation, seizures and cerebral oedema. Hesitancy of speech and deterioration in handwriting could be initial features, which can progress to aphasia, with both expressive and receptive components. Status epilepticus, fatal cerebral oedema and, occasionally, intracerebral hemorrhage could be the worst consequences. Cerebral MRI is requested to appropriately stage ICANS in order to choose the adequate therapeutic approach based on steroid use [100]. ICANS was reported in 18% of patients treated with ide-cel and 21% with cilta-cel, 3% and 9% of which were grade ≥3, respectively [90,101]. In the cilta-cel trial, delayed neurotoxicity was described as parkinsonism in seven patients, but further studies are needed to better understood the pathogenesis [102]. ICANS incidence was lower in trials with bispecific antibodies, always ≤5% and mostly grade 1–2, except for linvoseltamab, which caused 2% of grade 3 ICANS in the LINKER-MM1 trial [103]. Hemophagocytic lymphohistiocytosis (HLH)-like syndrome/immune effector cell-associated HLH-like syndrome (IEC-HS) is the third early complication of immunotherapies, recently described as an entity distinct from severe CRS. It is characterized as a hyperinflammatory syndrome with macrophage activation and HLH, cytopenia, hyperferritinemia, coagulopathy, hypofibrinogenemia and/or transaminitis [104]. Prior to infection, a longer CRS duration, grade ≥2 CRS and neurotoxicity were identified as risk factors [105]. Its incidence is very rare, only one patient treated with cilta-cel was reported. ASTCT working group consensus guidelines have been published for diagnosing, grading and treating IEC-HS, highlighting the need for a rapid clinical identification, initial treatment with anakinra and corticosteroids and escalation to dual therapy with the addition of ruxolitinib, etoposide and/or emapalumab [104]. Some peculiar toxicities should be considered for each immunotherapy. Unique toxicities with GPRC5D-targeted immunotherapies are related to its expression on keratinized tissues. Skin-related events (dry skin, eczema, pruritus, hyperpigmentation), nail-related events (discoloration, dystrophy, hypertrophy, onycholysis) and dysgeusia occurred in up to 70% of patients in the Phase 1 study of talquetamab [106]. The guidelines recommend supportive care, including emollient creams and oral rinses [107].

## 6. Infectious Complications Once Again in the Limelight

Infections have always represented a significant cause of morbidity and mortality in MM patients and, while, about 20 years ago, Augustson et al. reported that 45% of early mortalities were related to infections [108], ten years ago, Blimark et al. [109] found that MM patients had a significant 7-fold increased risk of developing any infection compared with matched controls, the risk being 11-fold higher in the first year. Several factors make MM patients susceptible to infections and they are patient-related such as age and comorbidities or disease- and treatment-related factors such as hypogammaglobulinemia, lymphopenia, impaired function of natural killer cells (NK) or neutropenia. In a post hoc analysis [110] of infections in 1347 patients (both young and elderly) enrolled in four clinical trials of the Spanish Myeloma Group and treated with PI- or IMiD-based regimens, the incidence of severe infections and the related mortality rate were 13.8% and 1.2%, respectively, within the first 6 months. A previous similar analysis of the FIRST trial [111], which included transplant-ineligible patients, showed a risk of grade ≥3 infections in patients receiving lenalidomide plus dexamethasone (Rd) of 22.6% in the first 18 months with 75% of infections occurring in the absence of neutropenia. Using the ECOG performance status, serum β2-microglobulin, lactate dehydrogenase and hemoglobin level, identified as factors significantly associated with early (in the first 4 months) grade ≥3 infections, the authors constructed a risk-scoring system able to stratify patients in high- and low-risk groups with significantly different rates of infection (24% vs. 7%, respectively, *p* < 0.0001). Data from randomized clinical trials [112] showed that patients receiving carfilzomib-based regimens were at a higher risk of serious infections if compared with the control group (pooled RR = 1.4, *p* = 0.0003), mainly respiratory tract infections. Regarding anti-CD38 mAb, patients treated with daratumumab had a 39% higher risk of developing pneumonia (RR = 1.39), with an RR of 1.38 for severe pneumonia, not translating, however, to a higher infection-related mortality rate [113]. Factors leading to higher risks of infections are not fully elucidated, and, while some studies emphasize the role of the decrease in absolute neutrophil count, lymphocyte count and NK count [114], in others no association between infection occurrence and NK cell count has been reported [115]. A recent consensus from an Italian expert panel provided recommendations for the prevention of infectious complications in patients treated with daratumumab [116].

Cytopenias and infections could be the most dangerous adverse events of novel immunotherapy, with early (<30 day from therapy) but also late onset (>30 days); in this case, increased bone marrow disease burden and longer CAR T cell persistence could be risk factors for prolonged cytopenia [117]. The most common cytopenia is neutropenia, which was globally 91% and 96% in CAR T cell therapy ide-cel and cilta-cel, respectively, with neutropenia grade ≥3 being 89% and 95% in the two groups, respectively [90,118]. Grade ≥3 neutropenia was 23–64% based on different bispecific antibodies in trials [85,86,87,89,119]. Infections occurred in the majority of patients treated with immunotherapies in pivotal trials, 22% and 20% grade ≥3 in KarMMa2 and CARTITUDE-1 ones, and from 9% to 45% in different bispecific antibodies trials, as reported above. Remarkably, non-BCMA-targeted BiTEs were associated with lower grade 3/4 neutropenia (25.3% vs. 39.2%) and grade 3/4 infections (11.9% vs. 30%) if compared with BCMA-targeted bispecific antibodies [120]. Real world experiences confirmed the high incidence of infections in patients treated with bispecific antibodies. Among 229 patients treated with this therapy in 14 French centers, 53% developed grade ≥3 infections, with a hospitalization rate of 56% and infection-related deaths of 9% [121]. Viral and bacterial infections are most common, with some fungal organisms observed [120,122,123,124,125]. A single-center observational study in patients with RRMM comparing infection incidence of patients treated with BCMA-targeting CAR T cells and bispecific antibodies in real world showed a more persistent infection risk and a higher incidence of severe infections in patients receiving BsAbs [126]. Infections have been associated with hypogammaglobulinemia, which was documented in 41% and 94% of patients in the KarMMa-2 and CARTITUDE-1 trials, respectively; whereas, in the BiTEs trials, hypogammaglobulinemia was <50% overall. In the study by Lancman et al. [127], prophylactic intravenous immunoglobulin (IVIg) led to a 90% reduction in grade 3–5 infections in patients receiving BCMA targeting BsAbs. Based on data from trials, consensus has been made about prophylaxis and therapeutic strategies for patients treated with immunotherapies. Antiviral prevention of HSV and VZV reactivation and antibacterial prophylaxis of Pneumocystis jirovecii are recommended for all patients, while specific antiviral/antifungal/antibacterial prophylaxes are reserved for selected patients (entecavir, azole, levofloxacin). Immunoglobulin replacement is recommended in the case of serum IgG ≤400 mg/dL, monitoring serum levels every 4 weeks. Influenza vaccine should be repeated annually and COVID-19 vaccine series repeated ≥3 months after CAR T cell therapy. If feasible, patients should be vaccinated prior to therapy, considering that patients receiving BCMA CAR T cell therapy experience a decline in pathogen-specific antibody titers to vaccinations [124]. G-CSF is recommended for active neutropenic infections, and it could be used if ANC < 1.0 × 10^9^/L, strongly recommended for ANC < 0.5 × 10^9^/L, especially if prolonged. A thrombopoietin-receptor agonist could be useful in the case of prolonged severe thrombocytopenia that persists beyond 30 days with high transfusion needs, but we have few data about it and this agent is not approved for this indication [83].

## 7. Conclusions and Future Directions

The management and treatment of patients with MM is becoming a complex and expensive process, with a relevant burden on the health system. In addition to the extreme biological heterogeneity of this hematological malignancy, requiring sophisticated methods to be studied, and the increasing availability of drugs and therapeutic approaches, we must remember that MM occurs mainly in patients over the age of 65 years, when concomitant comorbidities can represent a challenge to treatment decisions, especially in light of new generation immunotherapies as bispecific antibodies and CAR T cells that, although they can be administered at all ages, are burdened with potential serious toxicities. All this complexity requires the close collaboration of various health professionals, starting with the hematologist who diagnoses MM using the SLiM-CRAB criteria, although mainly SLiM criteria must be well evaluated and not slavishly applied, as above described. Biological characterization of MM represents a key way to identify not only prognostic factors (determining prognosis of patients with MM) but, more importantly, predictive factors essential for target therapy. While several staging systems such as ISS, R-ISS and more recently R2-ISS have been proposed (with the last two being an integration of the ISS stage system), predictive factors are very few, so targeted therapy is still a distant goal in the MM setting in contrast with solid tumors, in which it has revolutionized treatment and is successfully applied in various settings. However, with the precise assistance of artificial intelligence, the increasing number of genetic abnormalities, driver mutations and deregulated pathways identified by next-generation sequencing techniques will be used in the near future to construct predictive models to personalize treatment and to avoid drug resistance. MRD assessment at relevant time points during therapy, using not invasive methods, will probably become imperative since it will allow safe therapy discontinuation, avoiding overtreatment and occurrence of side effects. Several studies (MIDAS, PERSEUS, AURIGA, DRAMMATIC, OPTIMUMM) are exploring treatments tailored according to MRD response and their results could change the future management of MM. The introduction of novel immunotherapies such as CAR T cells and bispecific antibodies represent hope in triple-class refractory patients, i.e., those who develop refractoriness to PIs, IMiDS and anti-CD38 mAbs and for whom there have been no therapeutic options until recently. However, they are still very expensive and necessitate hospitalization to manage possible early severe toxicities such as CRS and ICANS and side effects not already known until their introduction. Increasing use of these novel immunotherapies (fixed duration or modified schedules of bispecific antibodies (currently being a continuous therapy)) could make these treatments quite manageable, thus mitigating infectious complications. On the other hand, the identification of predictive markers could help to identify patients with the highest likelihood of responding to immunotherapies, with significant cost reductions. Moreover, anticipation of these immunotherapies will allow to cure an increasing fraction of MM patients, providing that it will be possible to access the care.

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
