# Peer review of "The Challenging Approach to Multiple Myeloma: From Disease Diagnosis and Monitoring to Complications Management"

_cancers, 2024, doi:10.3390/cancers16122263_

Round 1

Reviewer 1 Report

Comments and Suggestions for Authors

Very nicely identified the complexity of care in multiple myeloma in the present era - but need to add about the ‘Interdisciplinary approach’, mentioned in the title, this can be expanded. Or you may consider a title modification.

Slim Criteria – please clarify -why you feel that early diagnosis and early treatment is bad? Does this critique fit with the review on multi-specialty care and increased complexity of management? May consider repositioning this in the paper. 

Comments on the Quality of English Language

Typo- in heading 

Infectious complications once again in the limelight

Author Response

Dear Reviewer 1,

  • We are trying to change our title, thanks for your suggestion
  • As for SLiM criteria: Ludwig et al. have recently demonstrated in a metanalysis that outcomes of patients with SLiM criteria in more recent papers  were much better that outcomes in previous papers, being very similar to MGUS outcomes, probably because in the past diagnostic techniques could identify syntomatic MM as early MM but too later. So SLim criteria shoul be rivisited and a more conservative approach should be applied when considering treatment initiation (Ludwig et al, EClinicalMedicine. 2023) 

Reviewer 2 Report

Comments and Suggestions for Authors

The introduction of novel agents as proteasome inhibitors, immunomodulatory agents and monoclonal antibodies as daratumumab and isatuximab resulted in significant improvement in survival either in newly diagnosed or in relapsed/refractory multiple myeloma (MM). In addition, new generation immunotherapies as CAR T cell therapy and bispecific antibodies recently showed to induce unprecedented rates of deep response in patients with very advanced MM. 

However, the use of these novel immunotherapies required clinicians to address with side effects not reported with previous treatments as cytokine release syndrome and neurotoxicity including immune effector cell associated neurotoxicity syndrome as well as with high rate of life-threatening infections. 

In this review MoreÌ€ et al. summarized the most recent topics on MM diagnostic and monitoring and described side effects that frequently occur during treatment with CAR T cells and bispecific antibodies. Therefore, this reviewer thinks that this review has a novelty and would be beneficial for readers in community.

Author Response

Dear Reviewer, thanks for your comment.

Reviewer 3 Report

Comments and Suggestions for Authors

please see file enclosed

Comments on the Quality of English Language

please see file

Author Response

Dear Reviewer, here the clarifications required:

  1. Varettoni et al study is a real life experience reporting data comparable with the Literature. In the review by Rosinol et al (Br J Haematol 2021) the incidence of extramedullary disease (not including paraskeletal plasmacytoma) ranges between 1.75% and 4.5% at diagnosis and 3.4%-10% at relapse and the Author points out that when PET/CT is systematically used at the time of diagnosis the reported incidence of EMD disease remains relatively low ranging from 2.4% to 10%. In a subsequent systematic review by Blade et al (Blood Cancer J, 2022), the reported incidence of EMD at diagnosis ranges from 0.5% to 4.8% while in relapsed/refractory MM it is 3.4% to 14%.
  2. We corrected, removing the term “novel”
  3. Ludwig et al (JCO 2023) evaluated whether the risk of progression of MM patients with SLiM CRAB criteria to classical CRAB criteria described in studies published after 2014 (when IMWG updated criteria for MM diagnosis was published) differs from that reported in studies leading to new criteria and published before 2014. In patients with ≥ 60% bone marrow plasma cells, time to progression (TTP) increased from 9.2 months in early studies compared to 30.3 months in later studies (three times longer as reported in our paper) whereas in patients with FLCratio ≥100 median TTP increased from 15.3 months to 48.1 months, respectively, with a 2-year risk of progression decreasing from 73% to 31.6% (50% lower 2-yr progression risk as reported in the paper). For patients with > 1 MRI focal lesion, no comparison between two time periods was possible since no studies were published after 2014.
  4. We corrected
  5. It's actually peremptorily and not preremptorily so we corrected. We just wanted to point out the well-documented reasons with which Mian et al, in response to Schavgoulidze et al study (Blood Cancer J, 2023), stated that 1) t(14;16) should continue to be considered high risk and 2) continue to be routinely included in test profiles.
  6. We corrected as suggested
  7. We modified the sentence on thrombopoietin-recepto agonist recommendation
  8. We refer to anti-CD38 and corrected
  9. We corrected the errors in the Table 1 and added GRIFFIN study
  10. In the Table 2 we added the main recommendation in clinical practice regaridin thecurrent and potential future laboratory esaminations and imaging methods for diagnosis and monitoring of MM
  11. We explained all abbreviations

Round 2

Reviewer 3 Report

Comments and Suggestions for Authors

The manuscript has significantly been improved by the changes made by the authors.

-Please specify that paraskeletal extramedullary disease was not included into estimation of EMD frequency

-the number and the sequence of references has been modified by the authors. Please check all references carefully.

-simple summary: MM is not really a `blood cancer´

-this statement is still not fully clear for me:...showed that in studies pub lished after 2014 patients with ≥ 60% bone marrow plasma cells or a FLC ratio ≥ 100 had an approximately three times longer TTP and 50% lower 2-yr progression risk compared to patients included in the earlier studies, suggesting that MM patients with the above SLiM findings may not be treated immediately but carefully followed up by stringent laboratory tests and medical examinations´. You compare MM patients with SLIM criteria before and after 2014? Why should OS be higher? both groups are untreated?

-Table 1. Please check: What is the meaning of ≥ CR? CR and stringent CR?

Author Response

  1. Paraskeletal extramedullary disease was not included into estimation of EMD frequency and we added this in the text
  2. We checked references carefully.
  3. We modified
  4. The study did not consider overall survival but time to progression to CRAB symptoms in newly diagnosed MM patients with SLiM finding (60% of clonal bone marrow plasma cells, involved/uninvolved serum free light chain ratio ≥100 and > 1 focal lesion on MRI studies). Since this time seems to be significantly longer in recent studies, particularly as regard 60% clonal bone marrow plasma cells or involved/uninvolved serum free light chain ratio ≥100, it is possible to treat patient not immediately. For example, in our experience, if at diagnosis a patient shows FLC ratio of 150 but 24 h proteinuria is less than 200 mg, we follow carefully patient but we do not treat him immediately
  5. ≥ CR is stringent CR